# Learning Sequential Acquisition Policies for Robot-Assisted Feeding

**Priya Sundaresan, Jiajun Wu, Dorsa Sadigh**
Stanford University
United States
priyasun@stanford.edu, {jiajunwu, dorsa}@cs.stanford.edu

**Abstract:** A robot providing mealtime assistance must perform specialized maneuvers with various utensils in order to pick up and feed a range of food items. Beyond these dexterous low-level skills, an assistive robot must also plan these strategies in sequence over a long horizon to clear a plate and complete a meal. Previous methods in robot-assisted feeding introduce highly specialized primitives for food handling without a means to compose them together. Meanwhile, existing approaches to long-horizon manipulation lack the flexibility to embed highly specialized primitives into their frameworks. We propose Visual Action Planning OveR Sequences (**VAPORS**), a framework for long-horizon food acquisition. **VAPORS** learns a policy for high-level action selection by leveraging learned latent plate dynamics in simulation. To carry out sequential plans in the real world, **VAPORS** delegates action execution to visually parameterized primitives. We validate our approach on complex real-world acquisition trials involving noodle acquisition and bimanual scooping of jelly beans. Across 38 plates, **VAPORS** acquires much more efficiently than baselines, generalizes across realistic plate variations such as toppings and sauces, and qualitatively appeals to user feeding preferences in a survey conducted across 49 individuals. Code, datasets, videos, and supplementary materials can be found on our website.

**Keywords:** Deformable Manipulation, Dexterous Manipulation

## 1 Introduction

Millions of people are impacted logistically, socially, and physically by the inability to eat independently due to upper mobility impairments or age and health-related changes [1, 2, 3]. Robot-assisted feeding has the potential to greatly improve the quality of life for these individuals while reducing caregiver burden. However, realizing a performant system in practice remains challenging. For instance, humans eat spaghetti noodles as shown in Fig. 1 using nuanced fork-twirling motions. Dishes like ramen require even more diverse strategies like scooping soup or acquiring meat and noodles. Thus, not only must an autonomous feeding system employ various utensils and strategies to handle different foods and quantities, but it must also operate over long horizons to finish a meal.

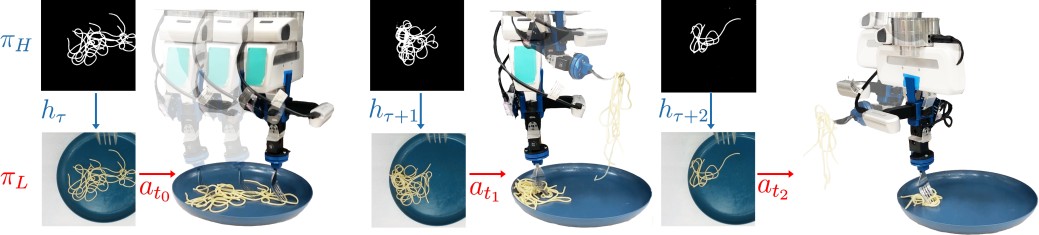

Figure 1: **Visual Action Planning OveR Sequences (VAPORS)** employs a high level policy $\pi_H$ to select amongst discrete manipulation strategies $h$, such as grouping and twirling, and a low-level vision-parameterized policy $\pi_L$ to execute these actions $a_t$ for long-horizon dexterous food acquisition.

7th Conference on Robot Learning (CoRL 2023), Atlanta, USA.

Prior assistive feeding work has focused on learning individual low-level vision-parameterized primitives for food manipulation. Examples include separate policies for skewering [4, 5, 6], scooping [7], bite transfer [8, 9, 10], cutting [11, 12, 13], and pushing food piles [14]. While highly specialized, these policies cannot reason over an extended horizon or make use of multiple strategies for more effective plate clearance. Humans, on the other hand, interleave *acquisition* and *rearrangement* actions with ease—pushing multiple peas together before scooping instead of painstakingly acquiring each individual pea or gathering noodles closer to each other before twirling with a fork. Replicating this long-horizon foresight in robotic feeding has yet to be demonstrated.

Recent work in skill-based reinforcement learning (RL) provides a natural way to model long-horizon manipulation sequences hierarchically. This entails first learning a high-level policy for composing skills [15, 16, 17], and then optionally inferring the parameters of low-level skills separately [18, 19, 20]. These approaches tend to favor learning from simulation to scale data collection [21], but current state-of-the-art simulators lack high-fidelity models for food deformation, visuals, and cutlery interaction. This complicates learning food manipulation policies in simulation and transferring them to real [13]. Existing hierarchical approaches also assume that the low-level skills come from a general-purpose library of primitives such as grasping and path planning [19, 22, 18, 15, 23], limiting their applicability to the food domain which requires highly specialized behaviors. Thus, we seek to find an appropriate layer of abstraction for feeding, which can leverage the benefits of (1) hierarchical planning for long-horizon manipulation and (2) vision-based primitives for fine-grained control. Our key insight is that learning from simulated experience only at a *high-level*, which need not capture the intricacies of food dynamics, and incorporating visual planning to instantiate *low-level* specialized primitives, yields a powerful approach to dexterous, multi-step food manipulation.

In this work, we present **VAPORS**: Visual Action Planning OveR Sequences, a unified framework for food manipulation. Our approach is decoupled into a high-level planner, which sequentially composes low-level primitives. We first learn a policy in simulation that models latent dynamics of plates from images. Specifically, we use segmented image observations as a representation space, which captures the distribution of food items and is transferable between simulation and reality for high-level plans. We train the policy using model-based RL with a reward that encourages both acquisition and rearrangement. Separately, we instantiate a library of specialized primitives in the real world from learned food pose estimation and segmentation. Finally, we use the learned high-level planner on segmented real food images to plan sequences of primitives for long-horizon acquisition.

We experimentally validate our approach on two real food manipulation tasks: robotic noodle acquisition and bimanual scooping. Across both real-world trials and a comprehensive user study of 49 users, **VAPORS** achieves the highest efficiency, plate clearance, and qualitative user ratings compared to heuristic and single-primitive baselines, all while generalizing to unseen plates.

## 2    Related Work

**Robot-Assisted Feeding.** Recently, a number of devices for mealtime assistance have become available on the market [24, 25], but are limited in functional reach due to reliance on pre-programmed trajectories or teleoperation by users. While *bite transfer* of a food item to a user's mouth is the eventual goal of autonomous feeding [9, 8, 10], we focus on *bite acquisition* as a primary initial step for downstream feeding. Prior work in bite acquisition demonstrates the effectiveness of visual planning for precise manipulation. Feng et al. [6], Gordon et al. [5, 26] and Sundaresan et al. [4] leverage bounding box localization, food pose estimation, and visual servoing to geometrically plan precise fork skewering motions. Similarly, Grannen et al. [7] and Suh and Tedrake [14] plan bimanual scooping and grouping actions, respectively, for segmented food piles. These works focus only on developing a specialized individual primitive for food manipulation. In isolation, this does not capture many long-horizon real-world feeding scenarios with multiple utensils and strategies.

**Long-Horizon Planning and Control.** Several recent frameworks tackle long-horizon manipulation by separating motion-level decision-making from sequential plans. Traditionally, task-and-motion-planning (TAMP) approaches tend to assume extensive domain knowledge including after-effects of actions and fixed task plans [27, 28, 29, 30, 31, 28]. In feeding, plate dynamics can be highly

uncertain, and state estimation is notoriously challenging, rendering these approaches ineffective. An alternative approach is model-based planning and control, with recent impressive results on complex tasks like dough manipulation [16, 32, 33]. This family of methods leverage learned environment dynamics over visual states like images [34, 35, 36, 37, 33], keypoints [38], or particle-based representations [16, 32] to sample and plan action sequences that maximize predicted rewards. However, these methods do not scale well to high-dimensional continuous action spaces such as that of food acquisition. To address this, hierarchical RL decouples policies into a high-level planner which selects amongst discrete but parameterized low-level primitives [39]. These works have demonstrated promising results on simulated long-horizon tabletop manipulation [18, 19, 15], but have yet to consider (1) real-world deployment beyond carefully controlled experimental setups, or (2) complex manipulation beyond commonplace primitives like pick-place, path-planning, and grasping. In contrast, we consider highly diverse plates requiring specialized primitives and tools.

**Learning and Control for Manipulation in the Real World.** A large body of robotics research focuses on learning real-world policies for manipulation either through sim-to-real transfer or exclusively from real interactions. With sufficient domain randomization, sim-to-real transfer has proven effective for tasks involving rigid objects or a limited set of deformable items like cloth, which state-of-the-art simulators support [40, 41, 42]. However, adapting these simulators to modeling food appearance and deformation is highly non-trivial. Meanwhile, learning exclusively from real data has been shown to work well in challenging domains such as semantic grasping [43] or cable untangling [44, 45, 46]. These approaches rely on state representations that are scalable to learn, such as descriptors learned from self-supervised interaction [43] or keypoints learned from a small amount of manually annotated images [47, 48, 49, 50]. In our setting, it is difficult to scale real-world data collection across the range of food shapes, appearances, and properties a robot may encounter. Self-supervised learning is also complicated due to resets and utensil interchange. We instead take a hybrid approach which takes advantage of simulation for modeling high-level plate dynamics from large-scale interactions, but leverages visual planning at the low level for precise real manipulation.

## 3 Problem Statement

We formalize the long-horizon food acquisition setting by considering an agent interacting in a finite-horizon Partially Observable Markov Decision Process (POMDP). This is defined by the tuple $(\mathcal{S}, \mathcal{O}, \mathcal{A}, \mathcal{T}, \mathcal{R}, T, \rho_0)$. We assume access to plate image observations $o_t \in \mathbb{R}_+^{W \times H \times C} = \mathcal{O}$ of unknown plate states $\mathcal{S}$, with the initial state distribution given by $\rho_0$. Here, $W$, $H$, and $C$ denote the image dimensions. $\mathcal{A}$ denotes the action space, and $\mathcal{T} : \mathcal{S} \times \mathcal{A} \to \mathcal{S}$ represents the unknown transition function mapping states and actions to future states. The time horizon $T$ denotes the discrete budget of actions to clear the plate and $\mathcal{R}(s, a)$ refers to the reward which measures progress towards plate clearance. Our goal is to learn a policy $\pi(a_t | o_t)$ that maximizes expected total return: $\mathbb{E}_{\pi, \rho_0, \mathcal{T}}[\sum_t R(s_t, a_t)]$, with $t \leq T$.

To do so, we decouple $\pi$ into separate high and low-level sub-policies. We assume access to $K$ discrete manipulation primitives $h^k$, $k \in \{1, \ldots, K\}$, and learn a high-level policy $\pi_H$ which selects amongst these primitives. Additionally, we learn a low-level policy $\pi_L$ which continuously parameterizes a selected primitive according to visual input. The components we aim to learn are summarized below, where $h^k$ denotes a discrete primitive type and $a_t$ denotes its continuous low-level instantiation:

$$\text{High-level policy} : \pi_H(h^k | o_{\leq t}, a_{\leq t-1}) \qquad \text{Low-level policy} : \pi_L(a_t | o_t, h^k)$$

We consider low-level actions $a_t$, parameterized by the position of the tip of a utensil $(x, y, z)$ and utensil roll and pitch $(\gamma, \beta)$. Here, $\beta = 0°$ corresponds to an untilted fork handle, for instance, and $\gamma = 180°$ corresponds to the fork tines being horizontal when viewed top-down (Fig. 2).

### 3.1 State-Action Representations

In this section, we outline the visual state and action representations which are at the core of our learning approach introduced in Section 4.

**Visual State Space.** Our approach makes use of RGB-D images and segmented plate observations, $I_t \in \mathbb{R}_+^{W \times H \times 3}$, $D_t \in \mathbb{R}^{W \times H}$, $M_t \in \mathbb{R}_+^{W \times H}$ at different levels of abstraction. We leverage binary

segmentation masks to capture the spread of food items on a plate, informing high-level planning with $\pi_H$, and RGB-D observations as input to $\pi_L$ which better capture fine geometric details of food.

**Action Parameterization.** We consider an agent that may either perform *acquisition* or *rearrangement* actions, parameterized below. Acquisition actions attempt to pick up a bite of food, and rearrangement actions consolidate items. For example, as a plate of noodles becomes more empty, the robot may need to employ a rearrangement action by pushing multiple strands together before twirling (acquiring) for a satisfactory bite size.

In *acquisition*, a robot with a utensil-mounted end-effector approaches the position $(x_d, y_d, z_d)$ in the workspace, and executes an acquisition motion parameterized by roll $\gamma$ and pitch $\beta$ (i.e. twirling, skewering, scooping, etc.). Here, $(x_d, y_d, z_d)$ denotes the *densest* location of the plate, where food is most closely packed to encourage a high-volume bite. Specifically, $a_{t,\text{acquis}} = (x_d, y_d, z_d, \gamma, \beta)$ (1).

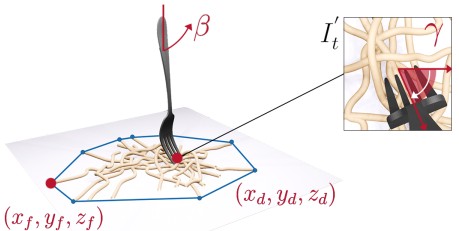

The intent of *rearrangement* is to bring food items from the sparsest plate region to the densest by pushing from $(x_f, y_f, z_f)$ ro $(x_d, y_d, z_d)$, while maintaining contact with the plate throughout. As this is a planar push, we simply orient the tool

Figure 2: **Action Parameterization:** We parameterize *acquisition* and *rearrangement* actions relative to the densest $(x_d, y_d, z_d)$ and furthest $(x_f, y_f, z_f)$ regions on the plate, as well as the utensil roll $\gamma$ and pitch $\beta$.

orthogonal to the push direction, such that $\gamma = \arctan\left(\frac{y_f - y_d}{x_f - x_d}\right)$, and is untilted ($\beta = 0°$): $a_{t,\text{rearrange}} = (x_d, y_d, z_d, x_f, y_f, z_f)$ (2).

## 4 VAPORS: Visual Action Planning OveR Sequences

Within the visual state and action space outlined in Section 3.1, we present our approach **VAPORS** for tackling long-horizon food acquisition. First, **VAPORS** learns a policy $\pi_H$, detailed in Section 4.1, to select amongst high-level strategies for long-horizon plate clearance via model-based planning. Finally, **VAPORS** learns a low-level policy $\pi_L$, which leverages visually-parameterized primitives to carry out generated sequential plans for real-world food acquisition detailed in Section 4.2.

### 4.1 Learning High-Level Plans from Simulation

Our goal is to first learn a policy $\pi_H$ for selecting amongst $K$ discrete acquisition or rearrangement strategies without concern for the low-level action parameters. To do so, we learn a latent dynamics model of the plate from segmented image observations, and instantiate $\pi_H(h^k | M_{\leq t}, a_{\leq t-1})$, $k \in \{1, \ldots, K\}$ with model-based planning over this learned dynamics model. In this section, $\tau$ denotes the running counter of high-level primitives executed so far, and $t$ denotes the current timestep.

**Simulator Overview.** We train $\pi_H$ entirely in simulation, where interactions can be collected at scale as opposed to the real world where manual plate resets and potential food waste are prohibitively expensive. As current simulators lack out-of-the-box support for many feeding scenarios, we develop a custom simulated food manipulation environment, visualized in Figure 3 in Blender 2.92 [40], further detailed in Appendix C.1. The simulator exposes RGB images $I_t$, binary food segmentation masks $M_t$, and food item positional states $s_t = \{(x_i, y_i, z_i)\}_{i \in (1, \ldots, N)}$.

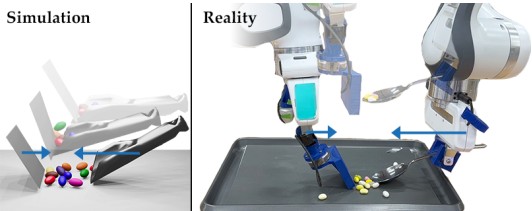

Figure 3: **Simulation vs. Real:** We visualize the task of bimanual scooping of jelly beans. Due to the sim-to-real gap, we merely leverage simulation to learn high-level food dynamics, and leave low-level action planning to real vision-parameterized primitives.

Using this information, we design rewards for food acquisition in terms of ground truth plate state and collect transitions to train $\pi_H$.

**Reward Design.** With access to a simulated testbed for feeding, we train $\pi_H$ to select amongst strategies via model-based reinforcement learning (RL). Our goal of efficient plate clearance can be

specified with a reward that incentivizes either (1) successfully picking up food, or (2) reducing the spread of items on a plate. Optimizing for the first objective alone might lead to plate clearance, but at a slow pace of taking low-volume bites. The second objective encourages rearrangement when the plate is sparse to aid downstream acquisition. Concretely, we express this as a weighted reward with tunable weight $\alpha \in [0, 1]$: $r_t = \alpha(\texttt{PICKUP GAIN}) + (1 - \alpha)(\texttt{COVERAGE LOSS})$ (3). Here, $\texttt{PICKUP}$ measures the quantity of food items picked up. $\texttt{COVERAGE}$ measures the spread of items on the plate, illustrated in blue in Fig. 2). We provide the details for computing both in Appendix C.2.

**Learning Latent Plate Dynamics.** With a means of measuring task progress via $r_t$ and access to plate observations $M_t$, we propose a model-based agent that learns plate dynamics from segmented observations and uses the learned model to plan actions that maximize reward. We achieve this by training a multi-headed latent dynamics model with the following (Fig. 4): (1) An *encoder* $q(z_t|M_{\leq t}, a_{\leq t-1})$ compressing high-dimensional segmented images $M_t$ to compressed latent states $z_t$, (2) A *transition function* over the latent states $p(z_\tau|z_{\tau-1}, h^k_{\tau-1})$ with which to imagine rollouts, and (3) A decoded *reward model* given by $p(r_t|z_t)$, such that at test time, we can sample action sequences and determine which maximize predicted rewards. We note that the transition function learns to predict high-level plate state changes between $\tau - 1$ and $\tau$ as a result of executing a primitive $h^k_\tau$, rather than between individual timesteps $t - 1$ and $t$ due to $a_t$.

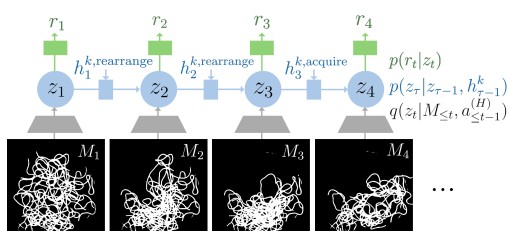

Figure 4: **Latent Plate Dynamics Model:** We learn a latent dynamics model of the plate comprised of an encoder $q$, transition model $p(z_\tau|z_{\tau-1}, h^k_{\tau-1})$, and a reward model $p(r_t|z_t)$. We use this model to select action sequences that maximize future rewards.

During training, we collect simulated transitions consisting of the masked image, high-level primitive, low-level action, and reward $\{(M_t, h^k_\tau, a_t, r_r)\}$. We train each head of this network using the objectives detailed in Appendix D.1.

We note that this approach is highly related to [37] with several crucial design choices. First, we learn plate dynamics over *segmented* image observations $M_t$ of food items on a plate, as opposed to raw RGB observations. This allows the dynamics model to attend to food items rather than the whole plate, provides an easily transferable representation between simulation and reality, and eases pressure for latent representations to capture irrelevant details in pixel space. Additionally, we learn a policy within an action space of discrete but continuously parameterized primitives as opposed to a high-dimensional space like joint-motor commands. This encourages actions that induce meaningful and perceptible plate changes likely to be encountered in downstream feeding.

**Model-Based Planning.** Once trained, we leverage the learned encoder, transition model, and reward model towards instantiating $\pi_H$ as an MPC-style planner with a receding $T$-step horizon. At timestep $t$, we enumerate all $K^T$ future candidate action sequences for the small library of primitives $K$. Conditioned on a history of observations $M_{1:t}$ and actions $a_{1:t-1}$, we imagine the future latent states $z_{\tau:\tau+T+1}$ under each action sequence $h^k_{\tau:\tau+T}$ via the transition function. Next, we predict decoded rewards according to the reward model $p(r_t|z_t)$ for each candidate sequence: $R = \sum_{i=\tau+1}^{\tau+T+1} \mathbb{E}\left[p(r_i|z_i)\right]$. Given the sequence of actions $(\hat{h}^k_\tau, \hat{h}^k_{\tau+1}, \ldots, \hat{h}^k_T)$ which maximizes predicted cumulative reward $R$, we take $\pi_H(M_{\leq t}, a_{\leq t-1}) = \hat{h}^k_\tau$, the first primitive in the predicted sequence. After executing this action, we replan with $\pi_H$, terminating when $\tau = T$. Details of the full planning pipeline, adapted from [37], are provided in Appendix D.2.

## 4.2 Visual Policies for Low-Level Real Manipulation

Our learned simulated task dynamics model from Section 4.1 relies on segmented images $M_t$ as an observation space and parameterized primitives as an action space. In this section, we describe the visual state estimation pipelines we use to instantiate our state-action representations on real data.

**Food Segmentation.** To define acquisition and rearrangement actions relative to the poses of food, we learn to segment food items on a plate as shown in Fig. 1. We learn a binary segmentation

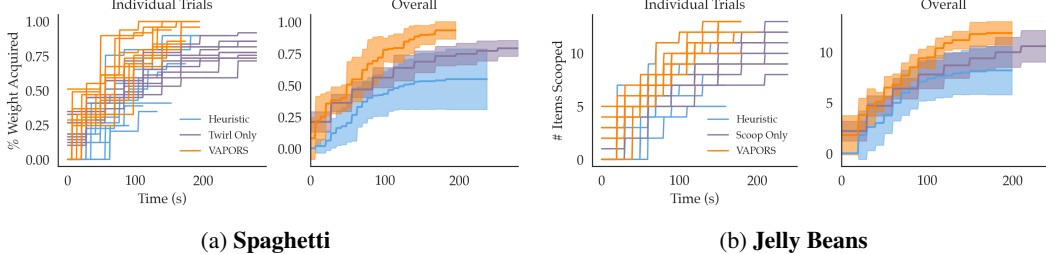

(a) **Spaghetti**             (b) **Jelly Beans**

Figure 5: Across 10 trials for spaghetti (a) and jelly bean (b) acquisition, we visualize the cumulative amount acquired across individual trials (left) and averaged overall (right). Shading denotes the standard error.

model $f_{\text{seg}} : \mathbb{R}_+^{W \times H \times 3} \to \mathbb{R}_+^{W \times H}$, where for a real image $I_t \in \mathbb{R}_+^{W \times H \times 3}$, $f_{\text{seg}}(I_t)$ yields a binary segmentation mask $\hat{M}_t$ which serves as input to $\pi_H$. To train $f_{\text{seg}}$, we require a paired dataset of real plate images and ground truth segmentation masks. However, manually labeling pixel-level segmentation annotations on images is a painstaking and time-consuming process for real plates of food. Instead, we use a self-supervised annotation process which starts by taking an image of an empty plate, gradually adding food items to the plate, and using the absolute frame difference between the empty plate image and current observation to obtain the food segmentation mask. We implement $f_{\text{seg}}$ as a fully convolutional FPN (Feature Pyramid Network) and train it according to the procedure detailed in Appendix D.3.

**Food Orientation.** Although segmentation provides a means to sense global *positional* information about food on the plate, we also care about precisely *orienting* a utensil with respect to the local geometry of a food item. For instance, using a fork to pick up a group of noodles requires orienting the fork tines opposite the grain of the strands. This is crucial to preventing slippage during twirling (Fig. 12), which tends to occur when the tines and strands run parallel. To address this, we also learn a network $f_{\text{ori}} : \mathbb{R}_+^{W' \times H' \times 3} \to \mathbb{R}$ mapping a local RGB crop of a food item of dimensions $W' \times H'$ to the desired roll orientation of the utensil $\gamma$. Prior work has shown that acquiring a food item orthogonal to its main principal axis, such as skewering a carrot against its length-wise axis rather than width-wise, can improve acquisition stability [4, 6]. Thus, we implement $f_{\text{ori}}$ as a fully convolutional network with a ResNet backbone and train it from a small amount of real food item crops (200), manually annotated with keypoints defining the principal food item axis as in [4].

**Action Instantiation.** With the visual state estimation pipelines $f_{\text{seg}}$ and $f_{\text{ori}}$ trained offline, we can instantiate $\pi_L(a_t | o_t, h^k)$ for real-world manipulation. Given an RGBD image observation $I_t, D_t$, we first infer the segmentation mask $\hat{M}_t = f_{\text{seg}}(I_t)$. Next, we query $\pi_H$ to obtain a selected primitive $\hat{h}^k = \pi_H(\hat{M}_{\leq t}, \hat{a}_{\leq t-1}^{(H)})$.

If $\hat{h}^k$ is an acquisition primitive, we instantiate the continuous action $a_{t,\text{acquis}}$ according to Eq. (1) by estimating the densest plate region $(\hat{x}_d, \hat{y}_d, \hat{z}_d)$ and utensil orientation $\hat{\gamma}$. To do so, we apply a standard 2D Gaussian kernel over $\hat{M}_t$ yielding $\hat{M}_t'$. This blurs the image such that high-density regions in the original segmentation mask remain saturated but sparse regions have lower intensity. From this, we take the 2D argmax $\hat{u}_d, \hat{v}_d = \arg\max_{(u,v) \in \hat{M}_t'} \hat{M}_t'[u, v]$ to be the densest pixel in the image, deprojected to a 3D location $(\hat{x}_d, \hat{y}_d, \hat{z}_d)$ via $D_t$ and known camera intrinsics. Given a food item crop centered at the densest pixel, $I_t'$ (Fig. 2) we also infer the utensil orientation with $\hat{\gamma} = f_{\text{ori}}(I_t')$. For a rearrangement primitive, we parameterize $a_{t,\text{rearrange}}$ according to Eq. (2). In addition to sensing the densest plate region, we sense the furthest region $(\hat{x}_f, \hat{y}_f, \hat{z}_f)$ by finding the lowest intensity pixel in $\hat{M}_t'$. This yields the following instantiations:

$$\pi_L(o_t, h^{k,\text{acquis}}) = (\hat{x}_d, \hat{y}_d, \hat{z}_d, \hat{\gamma}) \qquad \Big| \qquad \pi_L(o_t, h^{k,\text{rearrange}}) = (\hat{x}_d, \hat{y}_d, \hat{z}_d, \hat{x}_f, \hat{y}_f, \hat{z}_f)$$

Finally, **VAPORS** operates in a perception-action loop using $\pi_H$ to generate sequential plans and $\pi_L$ to execute them. The full algorithm can be found in Algorithm 1 of the Appendix.

## 5   Experiments

We seek to evaluate **VAPORS** ability to clear plates, by effectively leveraging diverse strategies and planning over long horizons. Thus, we compare against a single-strategy baseline with no

long-horizon reasoning and a multi-strategy approach that plans long-term actions heuristically rather than via learned plate dynamics. We consider two challenging real-world feeding scenarios to test the capabilities of **VAPORS** compared to other approaches: noodle acquisition and bimanual scooping.

**Experimental Setup:** In noodle acquisition (Fig. 10), a Franka robot with a wrist-mounted custom motorized fork and RGBD camera must decide amongst *twirling* (acquisition) or *grouping* (rearrangement) to clear a plate of noodles. In bimanual scooping (Fig. 11), two Franka robots operating from overhead RGBD cameras must select amongst *scooping* (acquisition) or *grouping* (rearrangement) to clear a plate of jelly beans. For both tasks, we consider a *half-full* initial plate distribution ( 50 g. noodles, 15 jelly beans) and a hard count of $\tau = 10$ actions for spaghetti and $\tau = 8$ actions for jelly beans, encouraging the acquisition of multiple items at once to finish a plate. For both tasks, we assume access to hand-eye calibration between the RGB-D camera and robot end-effector. In Appendix E we outline the hardware setup and control stack, low-level action instantiations, and training details for each task.

**Baselines:** *Acquire-Only* is identical to **VAPORS** in terms of $\pi_L$, but does not perform any long-horizon reasoning. Instead, at each timestep, this approach only acquires via twirling or scooping, with no rearrangement in between. *Heuristic* also utilizes $\pi_L$ in the same manner, but replaces $\pi_H$ with a naïve group-then-acquire strategy. This method senses the COVERAGE, as defined in Eq. (3), over $\hat{M}_t$ to heuristically determine when acquiring or rearrangement is appropriate. When the area exceeds a pre-defined threshold, the policy defaults to rearrangement and otherwise acquires.

**Plate Clearance Results:** We evaluate **VAPORS**, Acquire-Only, and Heuristic on clearing plates across 10 trials for each task (Fig. 5). We see that **VAPORS** achieves the most efficient and highest cumulative plate clearance. As expected, Acquire-Only optimizes only for acquisition in the current instant, without exploiting the benefits of grouping for a more substantial pickup of multiple items at once. Scooping one jelly bean at a time or attempting to twirl just a few strands of noodles repeatedly leads to the observed slow rate of overall clearance. Heuristic's greedy group-then-acquire approach plans based on detected coverage thresholds, which we find is brittle in practice especially for any artifacts in segmentation mask predictions. This naive metric also does not encourage acquiring any bite-sized piles that may form intermittently, but rather aims to amass everything into one large pile before acquiring. This delays acquisition gains and wastes the action budget.

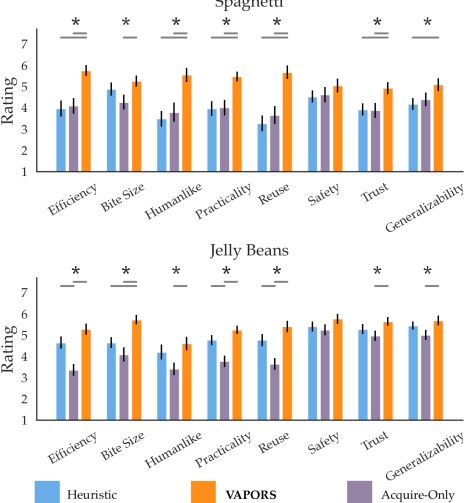

Figure 6: **Likert Ratings**: We administer a 7-point Likert survey to users after observing 10 trials per method. **VAPORS** elicits the most positive feedback across all criteria. '*' indicates statistical significance ($p < 0.05$).

**User Evaluation:** Additionally, we conducted a user study with 49 non-disabled participants (age range $27.0 \pm 9.5$, $46.9\%$ female and $53.1\%$ male) to gauge user preferences across methods. Of this pool, $77.6\%$ reported prior experience interacting with robots before, $75.5\%$ reported having fed someone before, and $28.6\%$ reported having been fed as an adult. We hypothesized: **H1.** *Compared to baselines, VAPORS use of multiple strategies and long horizon foresight will lead to more preferable feeding in terms of quantitative and qualitative metrics.*

We used a within-subjects design where we presented each participant with videos of all 10 plate clearance trials per each of the three methods, for either noodle acquisition or bimanual scooping. For each participant, we randomized the method order, the order of trials per method, and the food group. In the study, we ask participants to rate efficiency, bite size, similarity to human feeding, practicality, likelihood for reuse, safety, and generalization.

After watching all trials, we provided users with a 7-point Likert survey to assess these criteria (Fig. 6). **VAPORS** incurs the highest qualitative user ratings across criteria, compared to the Acquire-Only and Heuristic baselines, and with statistical significance for certain categories ($p < 0.05$, denoted '*'). Users noted that **VAPORS** "mimicked natural feeding," and "showed a capacity for clustering as the plate got more and more empty, which felt like a great and efficient approach," while Heuristic and Acquire-Only "seem like extreme policies, where [Acquire-Only] never

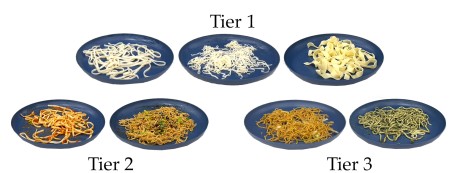

Figure 7: **Noodle Acquisition Tiers of Difficulty**: Tier 1 consists of plain noodle varieties: Dan Dan, Udon, and Pappardelle noodles. Tier 2 includes Tier 1 plates along with soy sauce, marinara sauce, and garnishes such as parsley or cilantro. Tier 3 plates include noodle dishes such as pesto pasta and chow mein ordered from DoorDash.

tries to cluster and [Heuristic] focus too much on making big piles." These results align with the hypothesis that **VAPORS**' use of multiple strategies and ability to reason over long horizons benefits a user's mealtime experience. We provide additional user study findings in Appendix E.

**Generalization Testing:** Finally, we stress-test **VAPORS'** generalization capabilities by experimenting with noodle dishes prepared with sauces and garnishes as well as ordered from DoorDash (Fig. 7). We conduct 18 additional trials of plate clearance on unseen plates, separated into three tiers of difficulty with 6 trials per tier. We summarize our findings in Table 1.

**VAPORS** achieves near full plate clearance for Tier 1 noodles, demonstrating generalization to noodle shapes and sizes (Table 1). While **VAPORS** is still able to make decent progress towards

| Tier | Description | % Cleared | *Failure Categorization* | | | | |
|------|-------------|-----------|---|---|---|---|---|
| | | | A | B | C | D | Failure Rate |
| 1 | Plain Noodles | $90\% \pm 6\%$ | 2 | 7 | 2 | 0 | 18% |
| 2 | Noodles w/ Sauce | $68\% \pm 16\%$ | 2 | 8 | 1 | 4 | 25% |
| 3 | DoorDash Noodles | $64\% \pm 13\%$ | 3 | 5 | 2 | 4 | 23% |

Table 1: **OOD Results and Categorization of Failure Modes:** (A) Misperception, (B) Wrong Action, (C) Imprecision, (D) Slip.

plate clearance in Tier 2, we observe the occurrence of more slip failures (D) and misplanned actions (A, B) due to the addition of sauce and distractor food items. Somewhat surprisingly, the performance gap between Tier 2 and Tier 3 is minimal, with **VAPORS** being able to clear well over half the noodles for a fully out-of-distribution plate. The main challenges include misperceiving cabbage for noodles in the chow mein, as well as dropping twirled noodles heavily coated in pesto or soy sauce (D) (Fig. 12). Regardless, **VAPORS** demonstrates promising signs of zero-shot generalization.

# 6  Discussion

We present **VAPORS**, which to our knowledge is the first framework to address the multi-step food acquisition problem in robot-assisted feeding. Our hybrid approach leverages simulation to learn to model high-level plate dynamics at scale, and uses visual pose estimation in order to perform dexterous maneuvers for complex low-level food pickup. We experimentally validate **VAPORS** on a complex suite of real-world food acquisition tasks such as noodle acquisition and bimanual scooping of beans. **VAPORS** demonstrates the ability to clear plates efficiently over non-learned baselines while appealing to the feeding preferences of real users.

**Limitations and Future Work.** The largest current limitation is a lack of user testing on individuals with mobility impairments that affect their ability to eat independently, discussed in detail in Appendix A. Additionally, although this work highlights promising initial results toward generalization across food variations such as shape, sauces, and toppings, we acknowledge that our library of low-level primitives is currently limited. One actionable future direction is expanding our library with prior work on skewering, cutting, and even toppling unstable items to tackle a more expansive set of plates. Our initial prototypes for dexterous food acquisition, such as the motorized fork, also open up interesting possibilities for future designs of dexterous interchangeable utensils which would enable rapid strategy switching. Currently, the system also executes primitives in an open-loop fashion, but we hope to use reactive control in the future to adapt online to slippage or imprecision.

## Acknowledgments

This work is in part supported by funds from NSF Awards 2132847, 2006388, 2218760, as well as Stanford HAI, the Office of Naval Research, AFOSR YIP FA9550-23-1-0127, and Ford. We thank Lorenzo Shaikewitz for designing the motorized fork used in this work which made real-world experimentation possible. We also thank Rajat Kumar Jenamani, Suneel Belkhale, Jennifer Grannen, Yuchen Cui, and Yilin Wu for their helpful feedback and suggestions. Priya Sundaresan is supported by an NSF GRFP.

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

# Learning Sequential Acquisition Policies
# for Robot-Assisted Feeding

Please refer to our website for videos, code, and supplementary material, as well as the 'Additional Experiments' page for supplementary ablations and a comparison to additional baselines. We first include a discussion around VAPORS' greatest current limitations in the larger context of robot-assisted feeding (Section A). In the subsequent sections, we also provide an overview of the main design choices behind **VAPORS** and a thorough overview of implementation and experimental details.

## A    Limitations and Risks

In this work, we evaluate VAPORS quantitatively in plate clearance experiments and qualitatively in user studies. However, the largest current limitation is a lack of comprehensive user testing on individuals with mobility impairments that affect their ability to eat independently. These individuals are not represented in the demographic of individuals surveyed as per Section 5, and we are actively working on expanding our pool of participants to include such users. We fully acknowledge that performing user studies with non-disabled participants is neither representative of the experience of individuals with difficulties eating, nor in any way a replacement for the feedback and insights these individuals could offer towards improving assistive feeding research. We also note that plate clearance success is not on its own an indicative metric of how useful or effective a system like VAPORS would be for assisting real users with eating difficulties. Extending VAPORS as part of a user-facing system for feeding would require several changes and considerations not explored in this work, such as taking into account different feeding preferences, mobility levels, involuntary head or body movements, comfort levels, and preferences surrounding shared versus full autonomy across different users. A full-stack feeding system would also need to carefully consider how to do in-mouth bite transfer safely, comfortably, and reliably. None of these considerations is exhaustive in any way; we merely see VAPORS as a step towards addressing challenges surrounding dexterity and long-horizon reasoning within bite acquisition, and we only speculate this having an impact for assistive feeding down the line. Towards this goal, we provide additional results below containing initial feedback from users with mobility impairments.

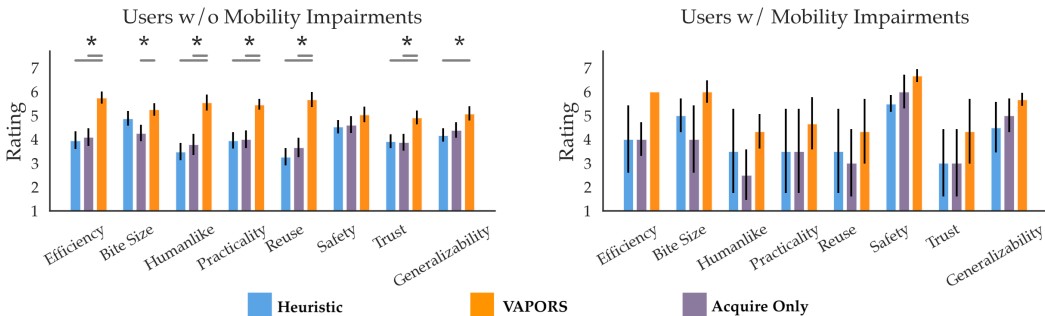

Figure 8: **Likert Ratings for Users with and without Mobility Impairments:** Error bars indicate standard error, and '*' indicates statistical siginficant findings via 1-way ANOVA testing. Users with and without mobility impairments favor VAPORS in terms of all criteria, but especially efficiency, bite size, and generalizability, suggesting the promise of our approach. For more user-facing considerations, such as practicality and whether users would reuse VAPORS given a choice, we find a small drop in average ratings between users without and with mobility impairments. In Section A, we discuss additional considerations to improve VAPORS in these aspects. We note that the larger error bars and lack of statistical significant findings for users with mobility impairments is due to the small sample size of users tested (3) as compared to 49 users without mobility impairments. We are actively working to increase this pool for larger-scale user testing in the future.

## B    Feedback from Users with Mobility Impairments

Although we have yet to physically evaluate VAPORS alongside users with mobility impairments, We seek to understand how these users perceive VAPORS currently. To do so, we conduct a user study via the same interface from Section 5 involving watching videos of food acquisition and providing Likert ratings as feedback. This study was approved by the Institutional Review Board of

Stanford University. In a survey sent out to users, we received three responses from participants (2 female, 1 male with ages in the range of 31-44). All participants reported difficulties and/or mobility impairments that affect their ability to eat independently, including a C5 quadriplegic injury with no finger or wrist function, and other impairments causing limited use of arms and hands.

We ask all users to evaluate the spaghetti acquisition trials from Section 5. In Figure 8, we visualize the average Likert ratings for users without versus with mobility impairments. We find that users with eating difficulties still rate VAPORS high in terms of efficiency, bite-size, and generalizability, and comparatively much higher than baselines. We do see that these users ratings for other considerations of a feeding system, such as practicality and reuse, are lower on average than users without mobility impairments, but VAPORS is still favored. Users reported that the twirling primitives occasionally struggled when there were very few strands of noodles left on the plate, or noodles were located at plate edges, which could lead to messy bites in practice.

Thus, while promising in terms of its algorithmic effectiveness, we fully acknowledge that VAPORS requires further refinement and ample testing to be a truly effective assistive feeding system, discussed in Appendix Section A. We further acknowledge that the sample size of users with mobility impairments is very small, and in the future, we hope to expand to a much larger pool of users to fully understand VAPORS' capabilities.

## C  Simulator Details

### C.1  Simulator Design

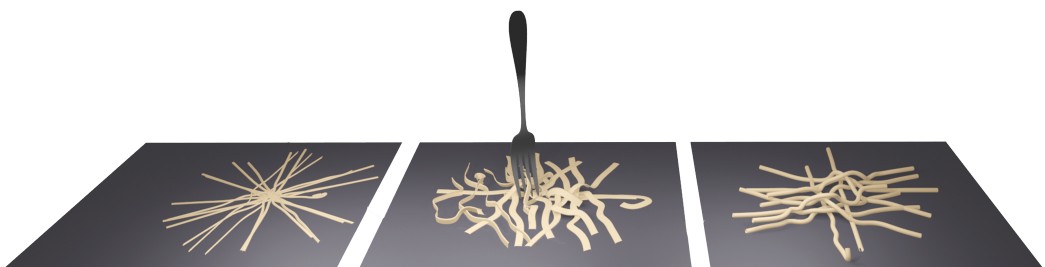

Figure 9: **Blender Food Simulation Environment:** We implement a custom food manipulation simulator in Blender 2.92 with an Open AI gym-style environment. The simulator supports softbody objects, such as noodles in different shape variations, as well as rigid, granular piles of items. We implement cutlery with arbitrary utensil meshes such as forks and spoons, and implement actions using the *keyframing* feature of Blender to control the position and orientation of a tool across frames.

We use Blender 2.92, a physics and rendering engine, to develop a custom feeding environment supporting deformable items, rigid items, and cutlery interactions. To instantiate deformable items like noodles (Fig. 9), we represent each item as a group of particles simulated with soft body physics. We treat granular piles of food such as jelly beans as separate rigid bodies. Additionally, we provide support for mesh-based utensils including a fork, spoon, and pusher tool, where we programatically keyframe the position and orientation of the tool across simulation frames to implement actions.

### C.2  Reward Design

In this section, we describe the implementation of the reward function given in Eq. (3). For a set of known food item states in simulation $s_t = \{(x_i, y_i, z_i)\}_{i \in (1,...,N)}$, PICKUP measures the quantity of food items picked up out of $N$ total items. We detect a picked up food item in simulation by thresholding the $z$ position of all items before and after an action, relative to plate height. Analogous to task progress metrics in cloth smoothing work [51, 52], we use COVERAGE to measure of spread of items on the plate. We compute this via the area of the convex hull of $\{(x_i, y_i)\}_{i \in (1,...,N)}$, depicted in Fig. 2, via the Scipy Python library.

## D  Details of Learning-Based Methods

### D.1  Latent Dynamics Training Details

We implement the latent plate dynamics model using the recurrent state space model from [37], with $64 \times 64$ input images and 30-dimensional diagonal Gaussian latent variables. This is a multi-headed

deep recurrent network comprised of a learned encoder, transition model, and reward model. We supervise each head of the network with the following objectives:

- For the *encoder* $q(z_t|M_{\leq t}, a_{\leq t-1})$, we use an auxiliary decoder head that upsamples latent variables $z_t$ to predicted images $\hat{M}_t$ and take the mean-squared error between $(\hat{M}_t, M_t)$ as a standard reconstruction objective. This encourages the learned latent representations to preserve the notion of food spread captured in segmented image observations.

- We supervise the *transition function* $p(z_\tau|z_{\tau-1}, h^k_{\tau-1})$ head using the KL-divergence for multi-step predictions as defined in [37].

- Finally, for the *reward model* given by $p(r_t|z_t)$, we take the mean-squared error between predicted rewards and ground truth rewards $(\hat{r}_t, r_t)$. This objective promotes accurately decoding rewards of future states to inform planning at test-time.

### D.2   Planning with Learned Dynamics Model

Once trained, we use an MPC-style loop to sample and plan actions that maximize predicted rewards under the learned reward model.

At time $\tau$, we can enumerate all $K^T$ future candidate action sequences for the small library of primitives $K$, where $T$ is the planning horizon. Conditioned on a history of observations $M_{1:t}$ and actions $a_{1:t-1}$, we imagine the future latent states $z_{\tau:\tau+T+1}$ under each action sequence $h^k_{\tau:\tau+T}$:

$$z_{t:t+T+1} \sim q(z_\tau|M_{1:t}, a_{1:t-1}) \prod_{i=\tau+1}^{\tau+T+1} p(z_i|z_{i-1}, h^k_{i-1}), \tag{4}$$

where $q(z_t|M_{\leq t}, a_{<t-1})$ is the learned encoder and $p(z_\tau|z_{\tau-1}, h^k_{\tau-1})$ is the learned transition model. Next, we predict decoded rewards according to the reward model $p(r_t|z_t)$ for each candidate sequence:

$$R = \sum_{i=\tau+1}^{i+T+1} \mathbb{E}\left[p(r_i|z_i)\right]. \tag{5}$$

Next, we select the sequence of actions $(\hat{h}^k_\tau, \hat{h}^k_{\tau+1}, \ldots, \hat{h}^k_T)$ which maximizes predicted cumulative reward $R$. The final step of the MPC planning loop is we take $\pi_H(M_{\leq t}, a_{\leq t-1}) = \hat{h}^k_\tau$, which is simply the first primitive in the predicted sequence. After executing this action, we replan with $\pi_H$, thus obtaining a second action and so on until $\tau = T$ (Algorithm 1).

---

**Algorithm 1** Planning with VAPORS

1: **for** $\tau \in \{1, \ldots, T\}$ **do**
2:    $I_t, D_t \leftarrow$ Get current RGBD image observation
3:    $\hat{M}_t = f_{\text{seg}}(I_t)$ // Infer segmentation mask
4:    $\hat{h}^k_\tau = \pi_H(\hat{M}_{1:t}, a_{1:t-1})$ // Select high-level action
5:    Execute $\pi_L(\hat{M}_t, \hat{h}^k_\tau)$

---

### D.3   Food Segmentation Training Details

**Self-Supervised Dataset Generation.**   To circumvent the painstaking process of pixel-level segmentation annotation for real food images, we design a self-supervised annotation procedure. First, we record a grayscale RGB image of an empty plate, $I_{\text{empty}} \in \mathbb{R}^{W \times H}_+$. Next, we manually place food items on the plate at random without changing the position of the plate, yielding a new grayscale observation $I_t$. Let $I_{\text{diff}} = |I_t - I_{\text{empty}}|$, the framewise absolute difference between the full and empty plate. We initialize the ground truth segmentation mask $M_t$ corresponding to $I_t$ as a 2D array of zeros, and then assign $M_t[I_{\text{diff}} > \texttt{THRESH}] = 1$. In practice, we find that $\texttt{THRESH} = 20$ reasonably separates the foreground from the background to detect food. With this procedure, we can scalably collect 280 paired RGB food images and segmentation masks in real within an hour

and a half of data collection. This includes plate resets, food placement, image capture, and offline background subtraction post-processing.

**Augmentation.** We augment this dataset 8X by randomizing the linear contrast, gamma contrast, Gaussian blur amount, saturation, additive Gaussian noise, translation, and rotation of each RGB image, applying only the affine component of these same transformations to the associated segmentation masks.

**Training Objective.** We train $f_{\text{seg}}$, implemented as a fully convolutional FPN (Feature Pyramid Network) using Dice loss:

$$\mathcal{L}_{\text{dice}} = 1 - \frac{2 \times \texttt{TP}}{2 \times \texttt{TP} + \texttt{FN} + \texttt{FP}} \tag{6}$$

This objective encourages high overlap between predicted and ground truth masks, as `TP, FN, FP` denote the number of pixel-level true positives, false negatives, and false positives in a prediction $\hat{M}_t$ compared to ground truth $M_t$.

# E  Experimental Details

## E.1  Noodle Acquisition Hardware Setup

Using a Franka Panda 7DoF robot, we aim to clear a plate of cooked noodles within a horizon of $T = 10$ actions. We fit the end-effector with a custom 3D-printed mount consisting of a RealSense D435 camera and a fork. To enable autonomous twirling and scooping capabilities, we extend the fork's range of motion via two servo motors (Dynamixel XC330-M288-T). We control the robot with a Cartesian impedance controller, where the programmable servos are integrated in the forward kinematics chain for positional control of the fork tip. The action space consists of either *group* (rearrangement) or *twirl* (acquisition) actions, instantiated according to the learned segmentation and pose estimation models detailed in Section 4.2.

A group action consolidates a sparsely distributed plate by sensing the furthest and densest points, $(\hat{x}_f, \hat{y}_f, \hat{z}_f)$ and $(\hat{x}_d, \hat{y}_d, \hat{z}_d)$, and executing a planar push from the furthest to densest point. In a twirl action, we infer the densest point and appropriate insertion angle $\hat{\gamma}$, roughly orthogonal to the grain of majority of the noodles. We use positional control to insert the fork into the densest noodle pile, and execute a fixed twirling motion by making two rotations at 6 radians per second. Finally, the fork scoops upward until nearly horizontal ($\beta = 80°$) and the robot brings the acquired noodles to a neutral position in the workspace.

For all trials, we use a non-slip plastic dinner plate, and mimic a bite successfully taken by a user after twirling by autonomously untwirling onto a discard plate.

## E.2  Bimanual Scooping Hardware Setup

We assume access to two Franka Panda robots, equipped with a pusher tool and a metal spoon, respectively, and an external RealSense D435 camera for perception. With this setup, we aim to acquire granular items on a plate using either *group* (rearrangement) or *scoop* (acquisition) actions, with a total action budget of $T = 8$ actions. In particular, we evaluate our system on the task of scooping jelly beans, but **VAPORS** is agnostic to the exact choice of food. Following the experimental setup of Grannen et al. [7], the spoon is mounted at an angle to the robot end-effector ($\beta = 30°$). The pusher is a concave 3D-printed tool intended to push piles of items into the spoon and maintain contact during lifting so as to prevent spillage.

Grouping actions are unimanual and use the pusher tool to push the sensed furthest item to the densest region on the tray. In a scoop action, we sense the densest pile and execute a parameterized motion in which the pusher and spoon move towards each other synchronously at a fixed $\gamma = 180°$. Once they arms are within a fixed threshold apart, the spoon scoops by tilting to $\beta = 80°$ and lifting to a neutral workspace position.

We conduct all trials on a standard cooking tray due to the enlarged manipulation workspace for two arms. To simulate a user's bite between actions, we manually discard the spoon contents after a scoop action.

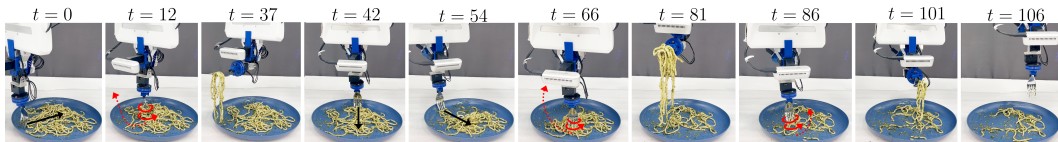

Figure 10: **Noodle Acquisition Rollout:** We visualize 6 actions performed by **VAPORS** on the task of clearing an initially half-full plate of Tier 3 noodles. As the distribution of noodles on the plate becomes sparse ($t = 0, 42, 54$), **VAPORS** employs grouping strategies (black) to push noodles in close proximity. Once consolidated, **VAPORS** employs twirling ($t = 12, 66, 86$), as shown in red, for efficient plate clearance, where $t$ denotes the clock time in seconds.

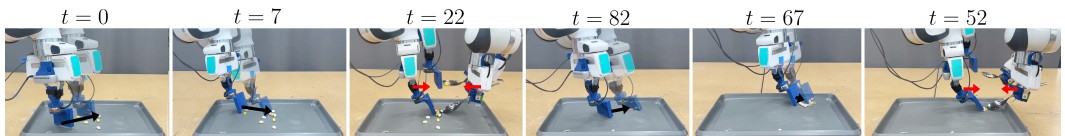

Figure 11: **Bimanual Scooping Rollout:** Using a bimanual setup with two Franka Emika Panda robots, **VAPORS** performs 6 actions consisting of grouping (black arrows) and scooping (red arrows) to acquire jelly beans on a tray. By grouping when the tray is sparse and acquiring when a bite-sized clump forms, **VAPORS** demonstrates efficient acquisition. The annotated timestamps denote clock time in seconds.

### E.3    Implementation Details

For each task, we use the following training procedures. We train $\pi_H$ on simulated segmentation observations of size $64 \times 64$ for $2,250$ update steps, where we collect 1 episode every $150$ update steps. We instantiate the reward as per Eq. (3) with $\alpha = 0.66$, and train each model using the Adam optimizer with with a learning rate of $10^{-3}$, $\epsilon = 10^{-4}$, and gradient clipping norm of $1000$ with batch size $B = 32$, based on the training procedure from [37]. Each model takes approximately 1 hour to train on an Nvidia RTX A4000 GPU. To instantiate $\pi_L$, we train $f_{\text{seg}}$ and $f_{\text{ori}}$ from real data. For segmentation, we collect 280 paired examples of images and binary segmentation masks using the self-supervised annotation process from Section 4.2, where we use cooked noodles of randomized shape and sauce variations as well as jelly beans of randomized colors. We augment each dataset 10X and train for 50 epochs, which takes approximately 3 hours on an NVIDIA GeForce RTX 2080 GPU. In order to instantiate the twirl primitive for noodle acquisition, we additionally train $f_{\text{ori}}$ to predict fork tine orientation $\gamma$ from 280 manually annotated crops of noodles as per Section 4.2, augmented 8X. The train time for $f_{\text{ori}}$ is approximately 1 hour on an NVIDIA GeForce RTX 2080 GPU. For deployment, we use an Intel NUC 7 for inference and robot control via a ROS 2-based control stack.

### E.4    Additional Experimental Results

In this section, we supplement the experimental findings from Section 5 with additional results.

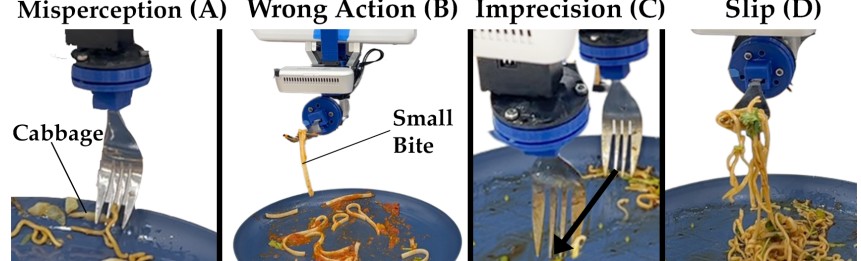

Figure 12: **VAPORS Failure Modes:** We illustrate the 4 most commonly observed failure modes with **VAPORS** on noodle acquisition. Misperception (A) occurs when $\pi_L$ erroneously senses vegetables, sauce, or plate glare as a noodle due to false positives with $f_{\text{ori}}$, leading to a misplanned action such as grouping in that region. Occasionally, $\pi_H$ may acquire when rearrangement is more appropriate, leading to a low-volume bite (B). In terms of action execution, food acquisition requires care so as to not miss food (C), as seen in the grouping motion which fails to group singular noodle strands due to system imprecision. Finally, slippage (D) can happen during acquisition with hard-to-model items such as those coated in sauce.

**Plate Clearance:** Fig. 10 and Fig. 11 visualize two rollouts of **VAPORS** on plate clearance. We note that visually, **VAPORS** tends to favor grouping as the plates become sparser and otherwise acquires when there is a reasonably sized bite available.

**VAPORS Failure Mode Categorization:** In addition to evaluating the percentage of the plate cleared, we observe the occurrence of a few failure modes, as depicted in Fig. 12. A *misplanned* action (A) can occur due to a perception error, such as accidentally perceiving sauce, a garnish, a vegetable, or plate specularity for noodles and erroneously grouping or twirling in that region. Alternatively, this can happen when (B) the robot twirls when grouping is more appropriate or vice versa. A *mis-executed* action failure occurs when (C) the fork fails to group or acquire due to system imprecision or (D) the noodles slip during acquisition due to sauce. In Table 1, we also report the per-action failure rate, computed as the total number of failures over the total number of actions (60 = 6 trials $\times$ $T = 10$).

**Qualitative User Study:** In the Likert survey administered to gauge user preferences across methods, we report in Fig. 6 the statistical findings which are significant. In Table 2, we indicate the specific margin of significance for each of the criteria, obtained via 1-way ANOVA testing.

Table 2: **1-Way ANOVA Statistically-Significant Findings** ($p$-value $< 0.05$)

| Criterion | Method 1 | Method 2 | $p$-value |
|---|---|---|---|
| Efficiency | Heuristic | VAPORS | 0.0004 |
| Efficiency | Acquire Only | VAPORS | 0.0010 |
| Bite Size | Acquire Only | VAPORS | 0.0318 |
| Humanlike | Heuristic | VAPORS | 0.0003 |
| Humanlike | Acquire Only | VAPORS | 0.0044 |
| Practicality | Heuristic | VAPORS | 0.0012 |
| Practicality | Acquire Only | VAPORS | 0.0025 |
| Reuse | Heuristic | VAPORS | 0.0000 |
| Reuse | Acquire Only | VAPORS | 0.0008 |
| Trust | Heuristic | VAPORS | 0.0124 |
| Trust | Acquire Only | VAPORS | 0.0198 |
| Generalizability | Heuristic | VAPORS | 0.0478 |

Table 3: **Noodle Acquisition**

| Criterion | Method 1 | Method 2 | $p$-value |
|---|---|---|---|
| Efficiency | Heuristic | Acquire Only | 0.0029 |
| Practicality | Heuristic | Acquire Only | 0.0091 |
| Reuse | Heuristic | Acquire Only | 0.0093 |
| Efficiency | Heuristic | Acquire Only | 0.0029 |
| Efficiency | Acquire Only | VAPORS | 0.0000 |
| Bite Size | Heuristic | VAPORS | 0.0029 |
| Bite Size | Acquire Only | VAPORS | 0.0002 |
| Humanlike | Acquire Only | VAPORS | 0.0094 |
| Practicality | Heuristic | Acquire Only | 0.0091 |
| Practicality | Acquire Only | VAPORS | 0.0001 |
| Reuse | Heuristic | Acquire Only | 0.0093 |
| Reuse | Acquire Only | VAPORS | 0.0001 |
| Trust | Acquire Only | VAPORS | 0.0438 |
| Generalizability | Acquire Only | VAPORS | 0.0481 |

Table 4: **Bimanual Scooping**

In addition to the user study outlined in Section 5, we administered a second part of the study, in randomized order to the first, in which users were asked to pick a preferred method for feeding in side-by-side comparisons of jelly bean acquisition trials. To control for the initial state of the jelly beans, we purposely arrange 16 beans into a $4 \times 4$ grid initially, and conduct two trials per method which are randomly selected for the comparisons. Although we would like to include an analogous side-by-side comparisons survey for noodle acquisition for completeness, we find in practice that controlling for the initial state of noodles is nontrivial due to their highly deformable nature and vast set of feasible initial configurations. This makes it difficult to present users with unbiased comparisons across methods.

Thus, for bimanual scooping, we presented all permutations of pairs of the three methods, for a total of six comparisons overall. Empirically, we find that **VAPORS** is the preferred method by a large margin compared to both baselines (Fig. 13).

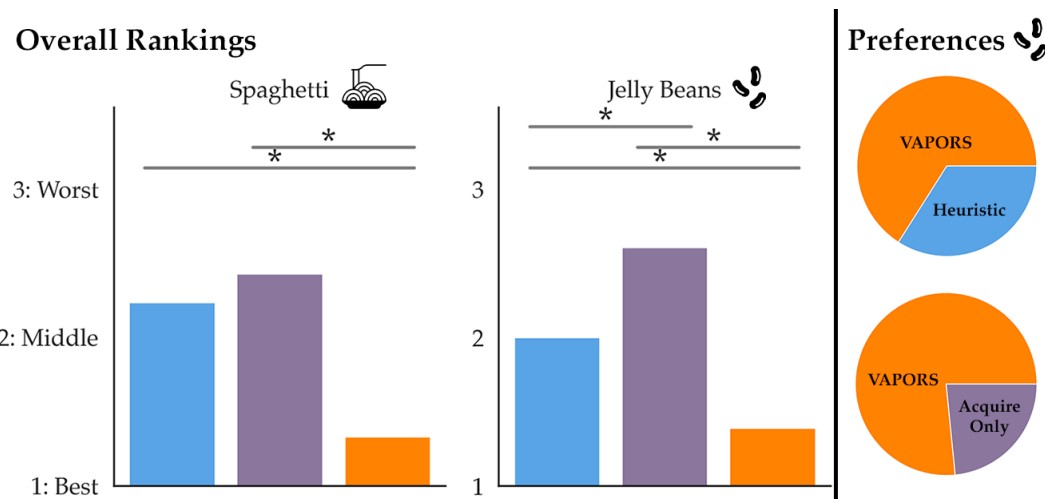

Figure 13: **Overall Ratings** Left: After observing all methods perform acquisition across 10 trials, we ask users to rank all three methods from most to least preferable. We find the **VAPORS** is most consistently ranked the best by a statistically significant margin ($p < 0.05$, denoted '*') compared to the baselines. Right: For jelly bean acquisition, we control for the initial state of the plate by arranging the beans in a $4 \times 4$ grid, and ask users to select their preferred method across 6 side by side acquisition videos of different methods. **VAPORS** is the preferred method by a large margin compared to Heuristic and Acquire-Only.

