# OpenReview forum: "Learning Sequential Acquisition Policies for Robot-Assisted Feeding"
_robot-learning.org/CoRL/2023/Conference — CoRL 2023 Poster_

### Official Review · Reviewer_cA3F · 2023-07-19

**Confidence:** 4
**Originality:** Very Good
**Technical Quality:** Good
**Clarity Of Presentation:** Very Good
**Impact:** 4

**Recommendation:**

Weak Accept: I recommend accepting the paper, but will not argue for my recommendation if the majority of other reviewers have a different opinion.

**Review:**

Strengths:
- The proposed system tackles a challenging robotics problem (manipulation of food) and shows that including long-horizon planning for when to group food and when to acquire a bite can improve effectiveness at that task
- The evaluation includes real-world foods (restaurant meals) as well as simpler food (jellybeans).

Weaknesses:
- The preference survey, conducted with non-disabled participants, does not tell the reader anything about the appropriateness of the method for assisted feeding for people with disabilities.  It is not appropriate to evaluate assistive technologies by asking nondisabled people to imagine being disabled. (see Cynthia L. Bennett and Daniela K. Rosner. 2019. The Promise of Empathy: Design, Disability, and Knowing the "Other". In Proceedings of the 2019 CHI Conference on Human Factors in Computing Systems (CHI '19). Association for Computing Machinery, New York, NY, USA, Paper 298, 1–13. https://doi.org/10.1145/3290605.3300528).  It would be better to leave this out or focus the study on how a co-diner would perceive the system.  If the perceptions of co-diners is not the point that the authors want to make, then they should conduct their study with disabled participants.  Overall, it is disappointing that the work has such high ecological validity in terms of the foods (ordered from real restaurants) and low ecological validity in terms of users, especially since the use of non-disabled proxies is not even listed as a limitation.
- Please clarify whether correction was used for the large number of statistical tests.


Minor points:
- In the first sentence, the "toll of meal preparation" is cited as a reason that people can't eat independently.  This is definitely not true (and the cited papers do not support this claim).

- I'm not sure that referring to sauce as adversarial is appropriate, given that sauce on noodles is a very normal configuration

**Quality Of The Limitations Section:**

Limitations are not well addressed

**Questions For Rebuttal:**

I would like the authors to address the ecological validity of the results with respect to intended users.  Ideally, I would like the authors to propose a plan for language to address the limitations with regard to interpreting the results and add nuance to their claims about user satisfaction.

**Robotics Focus:**

Sufficient demonstration on hardware

**Summary Of Paper:**

This paper presents a system for long-horizon planning for bite acquisition for a food manipulating robot intended to eventually be used in assistive feeding.

**Summary Of Recommendation:**

Overall, this paper has some exciting technical components but I am very concerned about the user study; it is not appropriate to evaluate assistive technologies with nondisabled users without careful attention to the interpretation of results and some discussion of how the results might change with disabled users.  I currently lean towards rejection, but this is highly dependent on whether the authors can provide a convincing plan in the rebuttal for providing this care and nuance.

Post rebuttal comment:
I am trusting that the authors will update their paper appropriately, but they have made an effort to address my main concerns (although having made an effort to contact users is not the same as having validated with those users).  I will raise my score to a weak accept.

---

### Official Review · Reviewer_iomA · 2023-07-20

**Confidence:** 4
**Originality:** Very Good
**Technical Quality:** Excellent
**Clarity Of Presentation:** Excellent
**Impact:** 4

**Recommendation:**

Strong Accept: I recommend accepting the paper and will argue for my recommendation even if other reviewers hold a different opinion.

**Review:**

**Strengths**
- The paper is expertly-written and was easy to follow. Numerous design choices are well-justified.
- The paper addresses an important problem and does a great job of motivating it and discussing the associated literature.
- Although not emphasized in the paper, I wonder if the modified simulator can be a valuable contribution to the research community focused on robot-assisted feeding.
- The experimental design is comprehensive. I particularly appreciate the effort to quantify and categorize complex food acquisition tasks and settings.
- The paper also includes a user study to demonstrate that users tend to prefer the proposed method over baselines.
- The authors have made their code and the dataset open-source.

**Weaknesses**
- While I appreciate the sound arguments in the paper pertaining to comparisons with existing methods, the paper lacks empirical comparisons with existing algorithms to substantiate the qualitative arguments.
- As the authors acknowledge, the set of primitives explored is rather limited. Adding new primitives will likely involve expert-design of parameterization and integration into the pipeline.
- It is not clear why the baselines are omitted from the generalization experiment.

**Quality Of The Limitations Section:**

Limitations are addressed clearly

**Questions For Rebuttal:**

Please address my questions and criticisms from the "Weaknesses and Suggestions" section of my review comments and let me know if there are any substantive misunderstandings on my part. Additionally, please address the following:
- In the generalization experiments, was the same policies (without any additional retraining/fine tuning) used for all 7 plates across 3 tiers?
- How was the coverage threshold for the heuristic baseline chosen?
- What specific statistical tests were performed? Were the associated assumptions validated?

**Robotics Focus:**

Sufficient demonstration on hardware

**Summary Of Paper:**

The paper presents a hierarchical algorithm to learn how to sequence food acquisition primitives associated with robot-assisted feeding. Specifically, a high-level planning policy is learned in simulation (to avoid sim-to-real gap) and a low-level visual policy is learned offline on real-world data to execute parameterized primitives.

**Summary Of Recommendation:**

The paper is well-written and addresses an impactful problem. The results conclusively demonstrate the effectiveness of the proposed method, including a user study. The paper lacks empirical comparisons with prior methods, but makes good qualitative arguments for how it improves upon them.

---

### Official Review · Reviewer_XrpJ · 2023-07-20

**Confidence:** 5
**Originality:** Fair
**Technical Quality:** Good
**Clarity Of Presentation:** Very Good
**Impact:** 3

**Recommendation:**

Weak Accept: I recommend accepting the paper, but will not argue for my recommendation if the majority of other reviewers have a different opinion.

**Review:**

Strengths:

The paper addresses interesting long-horizon robot-assisted feeding tasks -- robotic noodle acquisition and bimanual scooping. And the proposed method decouples the policy into separate high and low-level sub-policies.  The high-level part takes advantage of simulation for modeling high-level plate dynamics to leverage visual planning, and at the low level, the parameterized visual policy is executed for precise real manipulation. In general, the presentation for the method is clear and straightforward. Figures clearly express the algorithm step by step. There are sufficient quantitative results, baseline comparison and real-world execution videos look promising.

Weaknesses and suggestions:

1) It seems to me the two low-level primitives for the noodle task are all used during the execution which may be easy for the high-level dynamics model/planner to plan. Could you add an ablation study in sim by adding some redundant primitives here to test whether the high-level planner could ignore those redundant primitives and be robust to those "noises"
2) To further show the generalization power of the proposed method, it needs to have an example beyond clearing plates. For example, the planner should be able to generate a plan for making a certain pattern for noodles (e.g make the noodle into a circle).
3) One low-bound baseline is needed: treat random select low-level primitives as a high level policy and test the results
4) Another interesting baseline comparison: use the data trained for the dynamics model to train a primitive selector and treat the selector as a high-level policy and test the results

**Quality Of The Limitations Section:**

Limitations are addressed clearly

**Questions For Rebuttal:**

There are some minor points needed to be fixed:
1) In the abstract,
"Meanwhile, existing approaches to long-horizon manipulation lack the flexibility to embed highly specialized primitives into their frameworks."
Please consider modifying this to be more appropriate cause in the citations15-20 you mentioned, they have ways to embed.
2) It's better to put the planning objective function and algorithms back into the main paper for a better understanding

**Robotics Focus:**

Sufficient demonstration on hardware

**Summary Of Paper:**

This paper describes a framework for solving long horizon Acquisition Policies for Robot-Assisted Feeding. The proposed method Visual Action Planning OveR Sequences (VAPORS) uses a hierarchal structure where the high level selects discrete manipulation strategies and the low level executes vision-parametrized policies. Also, this paper shows results on two real food manipulation tasks: robotic noodle acquisition and bimanual scooping with high efficiency and plate clearance.

**Summary Of Recommendation:**

The paper shows an interesting solution for long horizon Acquisition Policies.  Though more details, experiments, and some modifications are needed to be added to show the validity of the method, I am leaning toward accepting this work.

---

### Author Response · Authors · 2023-08-12
**Overall Response to All Reviewers**

We thank all of the reviewers for their thoughtful comments and questions on our submission! In particular, we are encouraged that the reviewers appreciate our real-world evaluation efforts on a challenging manipulation domain, as well as the technical presentation of the paper. During the rebuttal period, we ask that the reviewers see the [Additional Experiments](https://sites.google.com/view/sequential-food-acquis/additional-experiments) tab of our website for new results based on the feedback we received.

**We first highlight concerns that were common amongst the reviewers, and later address each reviewer’s individual feedback point-by-point.**

* **Baseline Evaluation:**
  * Reviewers XrpJ and iomA suggest comparing VAPORS against additional baselines. First, it is worth mentioning that this is the first work to our knowledge which aims to tackle long-horizon food acquisition. As a result, we are not aware of a standard choice of baselines and we appreciate the reviewers suggestions towards establishing this precedent. The focus of many prior works [1,2,3] is developing a highly specialized, singular acquisition primitive like skewering or scooping, without regard for how to compose multiple available primitives. Therefore, we incorporate an acquire-only baseline for thoroughness. VAPORS, in contrast, explores long-horizon plate clearance by sequencing multiple available primitives. This raises the question: what is the best strategy to compose different primitives? The Heuristic baseline offers a very simple approach grounded in classical vision techniques, and we aim to see whether learned dynamics embeds a more sophisticated bias for VAPORS’ primitive selection. Along these lines, Reviewer 1 suggests comparing VAPORS against a high-level planner that chooses a primitive at random; we have since added this experiment in simulation. We show that VAPORS indeed outperforms this approach in terms of plate clearance and efficiency ([Additional Experiments](https://sites.google.com/view/sequential-food-acquis/additional-experiments): Random Baseline Experiments).
  * Reviewer iomA further requests “empirical comparisons with existing algorithms.” Since no particular method was indicated, we add a simulated experiment for spaghetti acquisition via end-to-end model-based RL ([Additional Experiments](https://sites.google.com/view/sequential-food-acquis/additional-experiments):  End-to-End Baseline). This experiment is intended to explore the use of a continuous action space over primitives altogether.  Based on the results, we see that the policy’s actions are undirected and completely lack emergent twirling or grouping behaviors. We hypothesize that this is due to the high-dimensional action space and lack of hierarchical planning, motivating VAPORS’ use of primitives.  Deploying such a policy to real is nontrivial due to safety concerns around utensil-plate contact, plate resets, and reward design without ground truth food item states. Alternatively, IL comes to mind as a possible alternative, but collecting high-quality, clean demonstrations for dexterous maneuvers like twirling remains challenging. Other potential alternatives are hierarchical RL approaches that learn to jointly compose primitives and infer the parameters of low-level primitives themselves. However, the types of dexterous primitives we consider have many dimensions (i.e. number of twirls, twirl angle, tilt angle, twirl position, velocity, etc. ). Attempting to infer all of these parameters from scratch is an ill-posed learning problem.
* **Primitives:**
  * Reviewer XrpJ and Reviewer iomA indicate that the set of primitives explored in this work is limited, and we fully acknowledge this point. However, we show that our primitives of scooping, twirling, and grouping can be instantiated from common and easily obtainable visual state representations such as segmentation masks and pose estimates. Prior work in primitive design for food acquisition has demonstrated that these visual state representations for food can be repurposed towards slicing [10], skewering [1,2], and bite-transfer [11]. We are currently working towards expanding VAPORS’ repertoire, and ([Additional Experiments](https://sites.google.com/view/sequential-food-acquis/additional-experiments): Expanding the Limited Set of Primitives) contains some preliminary videos.

---

> ### Author Response · Authors · 2023-08-12
> **Overall Response to all Reviewers (cont.)**
>
> **To summarize, the main updates since submission are:**
> * [Additional results](https://sites.google.com/view/sequential-food-acquis/additional-experiments) now available on the website:
>   * Two additional requested baseline results (end-to-end and random)
>   * Qualitative videos demonstrating an ongoing, expanded set of primitives
> * An updated plan regarding extending the user study to individuals with mobility impairments that affect eating, with over 25+ such users contacted since submission
> * Updated draft indicating all new writing changes in blue (attached as a pdf in each Rebuttal below):
>   * Disclaimer added to Limitations Section regarding how this work is not representative of the experience of users with mobility impairments, and how results may change with disabled users
>   * Minor writing changes throughout
>
> **Citations referenced throughout Rebuttal:**\
> [1] Feng, Ryan, et al. "Robot-assisted feeding: Generalizing skewering strategies across food items on a plate." Robotics Research: The 19th International Symposium ISRR. Cham: Springer International Publishing, 2022. \
> [2] Sundaresan, Priya, et. al. "Learning Visuo-Haptic Skewering Strategies for Robot-Assisted Feeding." 6th Annual Conference on Robot Learning. 2022. \
> [3] Grannen, Jennifer, et al. "Learning bimanual scooping policies for food acquisition." 6th Annual Conference on Robot Learning. 2022. \
> [4] Nasiriany, Soroush, et. al. "Augmenting reinforcement learning with behavior primitives for diverse manipulation tasks." 2022  International Conference on Robotics and Automation (ICRA). IEEE, 2022. \
> [5] Dalal, Murtaza,et. al. "Accelerating robotic reinforcement learning via parameterized action primitives." Advances in Neural Information Processing Systems 34 (2021): 21847-21859. \
> [6] Agia, Christopher, et al. "TAPS: Task-Agnostic Policy Sequencing." arXiv preprint arXiv:2210.12250 (2022). \
> [7] Hafner, Danijar, et al. "Learning latent dynamics for planning from pixels." International conference on machine learning. PMLR, 2019. \
> [8] Shi, Haochen, et al. "Robocraft: Learning to see, simulate, and shape elasto-plastic objects with graph networks." Robotics: Science and Systems (RSS), 2022 \
> [9] Zhou, Xingyi, et al. "Detecting twenty-thousand classes using image-level supervision." Computer Vision–ECCV 2022: 17th European Conference, Tel Aviv, Israel, October 23–27, 2022, Proceedings, Part IX. Cham: Springer Nature Switzerland, 2022. \
> [10] Sharma, Mohit, Kevin Zhang, and Oliver Kroemer. "Learning semantic embedding spaces for slicing vegetables." arXiv preprint arXiv:1904.00303 (2019). \
> [11] Shaikewitz, Lorenzo, et al. "In-Mouth Robotic Bite Transfer with Visual and Haptic Sensing." ICRA, 2023. \
> [12] Wang, Yixuan, et al. "Dynamic-Resolution Model Learning for Object Pile Manipulation." arXiv preprint arXiv:2306.16700 (2023). \
> [13] Wu, Jimmy, et al. "Tidybot: Personalized robot assistance with large language models." arXiv preprint arXiv:2305.05658 (2023). \
> [14] Bhattacharjee, Tapomayukh, et al. "Is more autonomy always better? exploring preferences of users with mobility impairments in robot-assisted feeding." Proceedings of the 2020 ACM/IEEE international conference on human-robot interaction. 2020. \
> [15] Bhattacharjee, Tapomayukh, et al. "Towards robotic feeding: Role of haptics in fork-based food manipulation." IEEE Robotics and Automation Letters 4.2 (2019): 1485-1492. \
> [16] Gordon, Ethan K., et al. "Leveraging post hoc context for faster learning in bandit settings with applications in robot-assisted feeding." 2021 IEEE International Conference on Robotics and Automation (ICRA). IEEE, 2021. \
> [17] https://meetobi.com/

---

### Decision · Program_Chairs · 2023-08-30

**Decision:**

Accept (Poster)

**Comment:**

The reviewers commend the paper for being well-written and organized. The proposed framework for long horizon food acquisition policies for robot-assisted feeding was well received by reviewers. I encourage the authors to further address any remaining points raised in the reviews for the final version of the paper.